# Effects of Intermittent Mild Cold Stimulation on mRNA Expression of Immunoglobulins, Cytokines, and Toll-Like Receptors in the Small Intestine of Broilers

**DOI:** 10.3390/ani10091492

**Published:** 2020-08-24

**Authors:** Shuang Li, Jianhong Li, Yanhong Liu, Chun Li, Runxiang Zhang, Jun Bao

**Affiliations:** 1College of Life Science, Northeast Agricultural University, Harbin 150030, China; 15804639630@163.com (S.L.); jhli@neau.edu.cn (J.L.); lyh_tre@126.com (Y.L.); lichun0917123@163.com (C.L.); 2Key Laboratory of Chicken Genetics and Breeding, Ministry of Agriculture and Rural Affairs, College of Animal Science and Technology, Northeast Agricultural University, Harbin 150030, China

**Keywords:** broilers, cold stimulation, immunoglobulins, cytokines, toll-like receptors, intestinal immunity

## Abstract

**Simple Summary:**

Cold stress has been associated with adverse effects on health and welfare of broilers. Whilst several studies have shown that long-term sustained and mild cold stimulation can improve immune function, little is known of the effects of intermittent cold stimulation on immune modulation in broilers. In this study, broilers were submitted to cold stimulation of 3 °C below than the usual rearing temperature during 3 and 6 h every two days during 43 days to explore its effect on the intestinal immunity. The findings confirm that appropriate mild cold stimulation has an overall positive influence on the intestinal immunity of broilers. The mild cold stimulation tested in this study is cost-effective and likely enhances overall health of broilers.

**Abstract:**

Appropriate cold stimulation can improve immune function and stress tolerance in broilers. In order to investigate the effect of intermittent mild cold stimulation on the intestinal immunity of broilers, 240 healthy one-day-old Ross 308 chickens were randomly divided into three groups: the control group (CC) housed in climatic chambers under usual rearing ambient temperature with a gradual 3.5 °C decrease per week; group II (C3) and group III (C6) to which cold stimulation at 3 °C below the temperature used in CC was applied every two days for 3 and 6 h, respectively, from day 15 to 35, and at the same temperature used in CC from day 35 to 43. The mRNA expression levels of immunoglobulins (*IgA* and *IgG*), cytokines (*IL2*, *IL6*, *IL8*, *IL17*, and *IFNγ*), and Toll-like receptors (*TLR2*, *TLR4*, *TLR5*, *TLR7*, and *TLR21*) were investigated in duodenum, jejunum, and ileum tissue samples on days 22, 29, 35, and 43. From day 15 to 35, mRNA expression of *IL2* and *IFNγ* was increased in the intestine of broilers. After one week of cold stimulation on day 43, mRNA levels of immunoglobulins, cytokines, and Toll-like receptors (TLRs) stabilized. Collectively, the findings indicate that cold stimulation at 3 °C below the usual rearing temperature had a positive impact on intestinal immunity of broilers.

## 1. Introduction

Cold stress impairs the performance of broilers by threatening health and welfare [1]. When the usual temperature is low as to induce cold stress in chickens, the immune system can become unbalanced [2]. Previous studies have found that appropriate cold stimulation can improve immune function and the ability to withstand cold stress and disease [1,3,4,5]. It has also been described that early cold conditioning improved late life thermotolerance in broilers challenged at 15 °C [6]. The intestine is an important responsive organ to cold stimulation due to its unique anatomical characteristics [7]. In fact, chronic cold stress has been shown to induce intestinal inflammation in rats and oxidative stress in quails [8,9], while acute cold stress induces intestinal injury, thus affecting intestinal immune function in broilers [10].

Immunoglobulins play an important role in maintaining intestinal immune function [11]. Certain antibodies produced in the intestinal mucosa can help increase host immune responses to resist endogenous and exogenous infections [12]. In particular, secretory *IgA* produced in the lamina propria is a vital protective molecule for the intestinal epithelium [13]. Moreover, proper cold stimulation has been shown to increase the expression of intestinal *IgA* and *IgG* to resist adverse effects [14].

Cytokines have an essential function in immunomodulation, participating in pro- and anti-inflammatory processes [15]. Dugué and Leppänen [16] suggested that cytokines can weaken or enhance the effect of adverse or advantageous environments on the human body, and that repeated cold stimulation increases tolerance to disease. Zhao et al. [10] showed that both acute and chronic cold stresses can increase the expression levels of interleukin-2 (*IL2*) and interleukin-17 (*IL17*) in the small intestine of chickens, causing tissue damage. Rhind et al. [17] found that long-term cold stimulation increased the levels of interleukin-6 (*IL6*) in the human serum, which limited the severity of inflammatory response. Previous research done by our group demonstrated that cold stress inhibited T helper 1 (Th1) cells response in broilers by reducing interferon gamma (*IFNγ*) and *IL2* levels, and long-term sustained stimulus at 3 °C below the usual temperature did not induce morphological injury, oxidative stress, or inflammation response in the ileum [18].

Intestinal immune function greatly influences organism overall health. Host immune response is elicited against invading microbial pathogens, and Toll-like receptors (TLRs) are constituents of a skillful system that recognize components of pathogenic microorganisms and regulate innate immunity [19]. *TLR2* and *TLR4* are involved in the intestinal immune response by identifying bacterial components [20]. Furthermore, TLRs signal activation of the NF-κB pathway and induce expression of *IL6* and interleukin-8 (*IL8*) to elicit innate and adaptive immune responses [21]. Paul et al. [22] demonstrated that mRNA expression levels of *TLR2*, *TLR4* and *TLR7*, as well as the activity of dendritic cells and macrophages, were reduced in the blood of Black Bengal goats submitted to long-term severe cold stress compared to goats housed in usual temperature in winter.

In previous studies, we have found that long-term sustained cold stimulation of 3 °C below the usual temperature improved immune function, enhanced adaptability to the environment, and conferred protection against injury of late acute cold stress in broilers [14,23]. However, little is known of the effect of a more energy-efficient strategy, i.e., the intermittent mild cold stimulation, on improving immunity in broilers. Therefore, the aim of this study was to evaluate the mRNA expression of immunoglobulins, cytokines and TLRs in the small intestine of broilers submitted to mild cold exposure to 3 °C below the usual rearing temperature for 3 and 6 h every two days.

## 2. Materials and Methods

### 2.1. Animals and Experimental Design

All experiments and procedures conducted in the present study have been previously approved by the Institutional Animal Care and Use Committee of the Northeast Agricultural University in Harbin, China (IACUCNEAU20150616). A total of 240 one-day-old Ross 308 male broilers were randomly divided into control group (CC), group II (C3), and group III (C6), each comprising 16 birds in 5 replicates. Broilers were battery caged in three climatic chambers for 43 days and given free access to water, a commercial starter diet (12.10 MJ/Kg metabolizable energy [ME], 21% crude protein [CP]) for the first three weeks, and a commercial growing-finishing diet (12.60 MJ/Kg ME, 19% CP) until the conclusion of experiments. Lighting regime was 24 h of light: 0 h of dark (24L:0D) for the first 3 days and 23 h of light:1 h of dark (23L:1D) from day 4 onwards. Relative humidity in the chambers was 60–70% from day 1 to 14 and 40–50% from day 15 to 43. Temperature scheme adopted in the experiments is shown in Figure 1. Birds in CC were held under a usual temperature scheme from day 1 to 43. Cold stimulation conditions consisted of 3 °C below the temperature used in CC. Commencing at 09:30 am on day 15, birds in C3 and C6 were exposed every two days to cold stimulation for 3 and 6 h, respectively, and standard CC temperature was subsequently restored in both groups. Cold stimulation ceased on day 35 at 17 °C and all three groups were managed at 20 °C from days 35 to 43. One bird from each replicate group was randomly selected and euthanized at 08:00 am on days 15, 22, 29, 36, and 43. Sections of the birds’ duodenum, jejunum, and ileum were excised, washed with 0.9% NaCl, immediately frozen in liquid nitrogen, and stored at −80 °C for subsequent RNA isolation.

### 2.2. Total RNA Extraction and Reverse Transcription

Duodenum, jejunum, and ileum tissue samples were ground to powder with a grinding rod using liquid nitrogen. Total cellular RNA was isolated from intestinal tissues using the RNAiso Plus kit (Takara, Japan) according to manufacturer’s instructions. Dried RNA pellets were re-suspended in 30 μL of DEPC-treated water. Total RNA concentration and purity were determined spectrophotometrically at 260/280 nm and RNA integrity was assessed by horizontal electrophoresis on a 1.5% agarose gel. Complementary DNA (cDNA) was synthesized using ReverTra Ace™ qPCR RT Master Mix with gDNA Remover (Toyobo, Osaka, Japan) following manufacturer’s instructions and stored at −80 °C.

### 2.3. Quantitative Real-Time PCR (qRT-PCR)

Genetic sequences available on GenBank for *β-actin*, *IgA*, *IgG*, *IL2*, *IL6*, *IL8*, *IL17*, *IFNγ*, *TLR2*, *TLR4*, *TLR5*, *TLR7* and *TLR21* of chicken were used as templates for primer design and subsequent synthesis by Sangon Biotech Co. Ltd. (Shanghai, China). Primer sequences are shown in Table 1. qRT-PCR reactions were performed in a LightCycler^®^ 96 (Roche, Switzerland) according to manufacturer’s instructions. SYBR^®^ Green I dye was incorporated in qRT-PCR reactions using THUNDERBIRD^®^ SYBR^®^ qPCR Mix kit (Toyobo). The 10 µL-final volume reaction mixture included: 1 μL of diluted cDNA, 0.3 μL of forward primer (10 mM), 0.3 μL of reverse primer (10 mM), 5 μL of SYBR Green I Master, and 3.4 mL of PCR grade water. qPCR conditions were as follows: initial heating at 95 °C for 1 min, followed by 40 cycles at 95 °C for 15 s and at 60 °C for 1 min. The melting curve showed a single peak for each PCR product. The house-keeping gene *β-actin* was used as an internal reference to determine gene expression. The relative abundance of mRNAs was calculated by the 2^−ΔΔCt^ method according to Schmittgen and Livak [24].

### 2.4. Statistical Analysis

Statistical analysis was performed using SPSS21 for Windows (SPSS Inc., Chicago, IL, USA). Two-way ANOVA and Duncan’s multiple comparison were used to analyze mRNA levels in small intestine tissue samples in response to cold stimulation and days. Intergroup and intra-group differences at given time points were analyzed by one-way ANOVA with Duncan’s multiple comparison. Results are expressed as mean ± standard error (SE) and probability value of less than 0.05 was significant (*p* < 0.05).

## 3. Results

### 3.1. Changes in mRNA Expression Levels of Immunoglobulins

The effect of cold stimulation experiments on mRNA expression levels of *IgA* and *IgG* in the duodenum, jejunum, and ileum of broilers are presented in Figure 2. mRNA levels of *IgA* and *IgG* in duodenum and jejunum of broilers were significantly affected by the treatments, age, and the interaction (*p <* 0.05). In the ileum, mRNA levels of *IgA* and *IgG* did not show significant differences (*p >* 0.05) except for the treatment-age interaction (*p <* 0.05).

According to Figure 2A, a significant increase in duodenal *IgA* mRNA expression comparing to CC (*p <* 0.05) was found in C3 on days 36 and 43, as well as in C6 on days 22 and 43. A significant increment in *IgA* mRNA levels was observed on day 43 in C3 and on day 36 in C6 (*p <* 0.05). Overall, cold stimulation applied to C3 significantly caused an increase in *IgA* mRNA levels over time (*p <* 0.05) but did not cause the same effect on C6 (*p >* 0.05). According to Figure 2D, considering duodenal *IgG* mRNA levels, it was found to be significantly enriched in C3 on days 22, 36, and 43, and in C6 on days 29 and 43 compared to CC (*p <* 0.05). Collectively, duodenal *IgA* and *IgG* mRNA levels (*p <* 0.05) were shown to be significantly increased in broilers of C3 and C6 groups a week after cold stimulation was removed (i.e., on day 43).

In jejunum tissue, comparing with the CC group, cold stimulation significantly increased *IgA* mRNA level on day 22 in C3 and C6 (*p <* 0.05, Figure 2B) with a subsequent decrease on day 36. No significant difference was found in *IgA* mRNA levels on days 29 and 43 among all three groups (*p >* 0.05). Interestingly, *IgG* mRNA levels (Figure 2E) were found to be significantly higher in C3 and C6 than in CC on day 22 (*p <* 0.05), but significantly lower in both groups compared to CC on day 29 (*p <* 0.05). Furthermore, *IgG* mRNA expression in C3 on day 43 was significantly higher compared to CC and C6 (*p <* 0.05), but levels found in the latter groups did not differ (*p >* 0.05).

In ileum, the effect of cold stimulation only determined a significant difference in *IgA* mRNA levels of broilers in C3 on day 29 (*p <* 0.05, Figure 2C), but not in C6 (*p >* 0.05) or CC. On the contrast, *IgG* mRNA levels were found to be decreased in C3 and C6 on day 22 (*p <* 0.05, Figure 2E), but no difference was found on other time points compared to CC (*p >* 0.05).

According to Figure 2F, a significant decrease in ileum *IgG* mRNA expression was found in C3 and C6 on day 22 compared to CC group.

As age of broilers progressed, the levels of *IgA* and *IgG* mRNA in both C3 and C6 showed a general increasing tendency compared to CC group.

### 3.2. Changes in mRNA Expression Levels of Cytokines in Duodenum

The results of mRNA expression of cytokines in duodenum of broilers are presented in Figure 3. *IL2* mRNA levels were significantly affected by the cold stimulation treatment, age, and treatment-age interaction (*p <* 0.05, Figure 3A). Compared to CC group, *IL2* mRNA levels in C3 were significantly higher on days 22, 36, and 43 (*p <* 0.05) and in C6 on days 36 and 43 (*p <* 0.05). As aging progressed, *IL2* mRNA expression levels in C3 and C6 showed a significant fluctuation.

*IL6* mRNA level (Figure 3B) was not affected by the cold stimulation treatment (P_T_ = 0.880) but by the age and treatment-age interaction (*p <* 0.05). Compared to CC, *IL6* mRNA levels in C3 and C6 were lower on day 29 and expressively higher on day 43 (*p <* 0.05). No difference was found among the three groups on other time points (*p >* 0.05).

When exposed to cold stimulation, *IL8* mRNA levels in the duodenum of broilers increased affected by the treatment, age, and treatment-age interaction (*p <* 0.05, Figure 3C). The *IL8* mRNA level in C3 group was significantly higher CC on days 22, 36, and 43 (*p <* 0.05). Moreover, *IL8* mRNA level was higher in C6 on day 29 and 36 (*p <* 0.05). With the progression of days, all three groups showed increased *IL8* mRNA levels on days 36 and 43 than on days 22 and 29 (*p <* 0.05).

Interestingly, cold stimulation treatment did not induce alterations in duodenal *IL17* (P_T_ = 0.486, Figure 3D) but they were affected by the age and treatment-age interaction (*p <* 0.05). Compared to CC group, the *IL17* mRNA was significantly lower in C6 on day 22 and 43. The *IL17* mRNA level in C6 was significantly higher on day 29 compared to CC group and in the case of C3 group the *IL17* mRNA was significantly lower on day 43 and higher on 29 day compared to CC group. No consistent trend was verified for mRNA levels of *IL17* over time, although significant differences were found in some time points.

*IFNγ* mRNA level was not affected by the cold stimulation treatment (P_T_ = 0.085, Figure 3E) but *IFNγ* mRNA level was by the age and treatment-age interaction (*p <* 0.05). *IFNγ* mRNA level was significantly higher in C6 compared to CC on days 36 (*p <* 0.05), but significantly lower on day 22 (*p <* 0.05). As days progressed, *IFNγ* mRNA levels in all groups fluctuated.

### 3.3. Changes in mRNA Expression Levels of Cytokines in Jejunum

Changes in mRNA levels of cytokines in the jejunum of broilers are presented in Figure 4. mRNA levels of *IL2*, *IL6*, *IL8*, *IL17*, and *IFNγ* were significantly affected by the cold stimulation treatment, days of age and treatment-age interaction (*p <* 0.05). Overall, mRNA expression of *IL2*, *IL6*, *IL8*, *IL17* was significantly higher on days 22 and 29 in groups exposed to cold stimulation.

Compared to CC, *IL2* mRNA expression in C3 was significantly higher on days 22 and 29 (*p <* 0.05, Figure 4A) and in C6 on day 36 (*p <* 0.05). As days progressed, *IL2* mRNA levels in C3 decreased between days 29 to 36, and in C6 increased on day 29 and then it also remains at the same level on days 36 and 43.

*IL6* mRNA level was found to be significantly increased in C3 on days 22 and 43 (*p <* 0.05), and showed no consistent trend across days of age in all groups (Figure 4B).

*IL8* mRNA expression in C3 showed a significant increase compared to CC on days 22 and 29 (*p <* 0.05, Figure 4C), but no difference in *IL8* mRNA levels was found between C6 and CC on day 29, 36, and 43 (*p >* 0.05). No trend was found in *IL8* mRNA expression among all groups with age progression.

*IL17* mRNA levels in C3 and C6 showed a significant increase compared to CC only on day 22 (*p <* 0.05, Figure 4D). A significant higher *IL17* mRNA level was in all groups on day 22 than on days 29, 36, and 43, respectively (*p <* 0.05).

*IFNγ* mRNA levels were significantly higher in C6 compared to C3 and CC on days 36 and 43, but lower than C3 and CC groups on day 22 (*p <* 0.05, Figure 4E), tending to increase in C6 as days progressed. No specific trend in *IFNγ* mRNA levels was found in C3 and CC over time.

### 3.4. Changes in mRNA Expression Levels of Cytokines in Ileum

The results of mRNA levels of cytokines in ileum are shown in Figure 5. mRNA levels of *IL2*, *IL6*, and *IL8* were significantly affected by the cold stimulation treatment, as seen throughout age and treatment-age interaction (*p <* 0.05). *IL17* mRNA level was also affected by treatment-age interaction (*p <* 0.05).

*IL2* mRNA levels in C3 showed an increasing tendency as days progressed and were significantly higher on days 22 and 29 compared to CC (*p <* 0.05, Figure 5A). In C6, *IL2* mRNA expression was lower on day 22 and higher on days 29 and 36 compared to CC group (*p <* 0.05). On day 43, there were no differences in *IL2* mRNA levels among C3, C6, and CC (*p >* 0.05).

*IL6* mRNA level was significantly increased in C3 on day 43 and decreased in C6 on day 29 compared to CC (*p <* 0.05, Figure 5B), and no significant trend for *IL6* mRNA levels was found in C3 and C6 over time.

*IL8* mRNA levels in C3 and C6 were significantly lower on day 22 than those in CC (*p <* 0.05, Figure 5C). *IL8* mRNA level was higher in C3 on days 29 and 36 but lower in C6 on day 43 compared to CC (*p <* 0.05). No varying trend in *IL8* mRNA expression was found over time in all groups.

*IL17* mRNA level (Figure 5D) was found to be lower on day 22 and higher on day 29 in C3 than CC, and *IFNγ* mRNA expression was significantly enriched in C3 (Figure 5E) than in CC on day 22 (*p <* 0.05). No difference was found in the mRNA levels of *IL17* and *IFNγ* (*p >* 0.05) among all three groups on other time points.

### 3.5. Changes in mRNA Expression Levels of TLRs in Duodenum

The results of mRNA expression levels of TLRs in duodenum are shown in Figure 6. mRNA levels of *TLR2*, *TLR4*, *TLR7*, and *TLR21* were significantly affected by cold stimulation treatment and age (*p <* 0.05), but *TLR5* mRNA levels were only significantly affected by cold stimulation treatment (*p <* 0.05). mRNA levels of *TLR4*, *TLR5*, and *TLR21* were also significantly affected by the treatment-age interaction (*p <* 0.05).

*TLR2* mRNA levels in C6 on days 22, 29, and 36 were significantly higher than in C3 and CC (*p <* 0.05, Figure 6A). *TLR2* mRNA expression in C3 decreased from days 22 to 29 and then increased on day 36, but no trend for *TLR2* mRNA expression was found in C6 and CC with age progression.

*TLR4* mRNA level was found to be significantly higher in C3 compared to C6 and CC on days 22, 36, and 43 (*p <* 0.05, Figure 6B), and no difference was found between C6 and CC on day 43 (*p >* 0.05). *TLR4* mRNA level showed no trend across days of age in all groups.

*TLR5* mRNA level was significantly enriched in C3 on day 43 (*p <* 0.05, Figure 6C) compared to C6 and CC (*p >* 0.05). No noticeable trend of *TLR5* mRNA levels was found over time in the tested groups.

*TLR7* mRNA levels were found to be affected by the cold stimulation treatment, and significantly higher *TLR7* mRNA expression was found in C3 and C6 on days 22, 29, 36, and 43 (*p <* 0.05, Figure 6D). *TLR7* mRNA levels in all groups tended to decrease between days 22 to 29 and then increase on day 43.

Comparing with CC, *TLR21* mRNA expressions in C3 and C6 were significantly lower on days 22 and 29 (*p <* 0.05, Figure 6E). When birds were submitted to cold exposure, *TLR21* mRNA level in CC significantly decreased as age progressed (*p <* 0.05), and then significant increased on day 43 (*p <* 0.05), but the expression of *TLR21* in C3 and C6 did not show the trend with age (*p >* 0.05).

### 3.6. Changes in mRNA Expression Levels of TLRs in Jejunum

The fluctuations in mRNA levels of TLRs in jejunum of broilers are shown in Figure 7. mRNA levels of *TLR2*, *TLR4*, *TLR5*, *TLR7*, and *TLR21* were significantly affected by the cold stimulation treatment, age, and the interaction between treatment and age (*p <* 0.05).

*TLR2* mRNA levels in C3 and C6 on days 22, 29, and 43 were significantly higher than in CC (*p <* 0.05, Figure 7A), while expression in C3 on day 36 was lower than that in C6 and CC (*p <* 0.05). No trend was found for *TLR2* mRNA expression over age among tested groups.

*TLR4* mRNA expression was lower on days 22, 36, and 43 in C3 compared to CC (*p <* 0.05, Figure 7B), while in C6 group was higher on day 43 and lower on 22d than CC group (*p <* 0.05). All groups showed a higher *TLR4* mRNA level on 36d than other detected time points, respectively (*p <* 0.05).

*TLR5* mRNA level was found to be significantly higher in C6 compared to C3 and CC on day 22 (*p <* 0.05, Figure 7C). *TLR5* mRNA level was significantly lower in C3 compared to C6 and CC on day 36. There was no regular trend found for *TLR5* mRNA level over days.

After one week of ceasing exposure to cold (day 43), *TLR7* mRNA levels in C3 and C6 were higher than in CC (*p <* 0.05, Figure 7D). Interestingly, a significant increasing trend in *TLR7* mRNA expression was observed in C6 (*p <* 0.05).

When cold stimulation exposure completed one week (day 22), *TLR21* mRNA level (Figure 7E) was found to be significantly increased in C6 (*p <* 0.05) compared to C3 and CC (*p <* 0.05). All groups showed increased *TLR21* mRNA levels on day 22, which progressively decreased in subsequent time points (*p <* 0.05).

### 3.7. Changes in mRNA Expression Levels of TLRs in Ileum

Levels of mRNA expression of TLRs in ileum are presented in Figure 8. *TLR2* mRNA expression (Figure 8A) was not affected by cold stimulation (P_T_ = 0.443) but by age and treatment-age interaction (*p <* 0.05). *TLR2* mRNA levels in C6 showed significant lower compared to CC on days 22 and 43 (p < 0.05), and higher on day 36 (*p* < 0.05), whereas no difference in *TLR2* mRNA levels was found between C3 and CC on days 29, 36, and 43 (*p >* 0.05). Moreover, *TLR2* mRNA levels in C3 were significantly lower compared to CC on days 22 (*p* < 0.05). *TLR2* mRNA levels in C6 and CC tended to be decreased on days 22 and 29, and increased as days progressed.

*TLR4* mRNA expression was affected by treatment, age, and treatment-age interaction (*p <* 0.05, Figure 8B). Significant increase in *TLR4* mRNA expression was found in C3 compared to CC on day 43 (*p <* 0.05), and the expression in C3 and CC tended to increase over time.

*TLR5* mRNA level was not affected by treatment (P_T_ = 0.060) but by age and treatment-age interaction (*p <* 0.05, Figure 8C). *TLR5* mRNA level in C6 was higher than in C3 on day 36 and in CC on day 22 (*p <* 0.05). As days progressed, *TLR5* mRNA levels in all groups fluctuated.

Application of cold stimulation treatment and age were found to not affect *TLR7* mRNA levels (*p >* 0.05, Figure 8D) but treatment-age interaction showed a significant effect (*p <* 0.05). *TLR7* mRNA levels in C3 and CC were lower than in C6 on day 36. With age, *TLR7* mRNA levels in CC tended to increase.

*TLR21* level was not affected by age (*p <* 0.05, Figure 8E) but by treatment and treatment-age interaction (*p >* 0.05). *TLR21* mRNA levels in C3 and CC were higher than in C6 on day 29. *TLR21* mRNA level in C6 and CC also fluctuated over time.

## 4. Discussion

Cold stress induces a pronounced suppression of humoral and cellular immune responses [25,26]. Proper cold stimulation can improve immune function and enhance resistance to diseases [3,27]. It has been demonstrated that cold stimulation enhances or suppresses cell-mediated immune response, depending on the duration and degree of exposure to the usual temperature [28,29,30]. In previous studies, we showed that sustained long-term cold stimulation conditions (3 °C below the usual rearing temperature) applied from eight days of age did not induce damage in broilers [23,31]. In the present study, mRNA expression levels of immune-related genes of broilers was investigated after an intermittent cold stimulation at 3 °C below the usual temperature still can modulate the intestinal immunity of broilers.

Immunoglobulins are a major component of the immune system, playing an important role in the intestinal immune function [32]. *IgA*, the most abundant immunoglobulin in the intestinal tract, is a key barrier of mucosal immunity [33]. Carr et al. [34] indicated that a proper cold stimulus can regulate autoimmune function by mediating immunoglobulin production, who found that the expression of intestinal *IgG* and *IgA* was increased when mice were placed in a −20°C freezer 20 min per day. Zhao et al. [10] showed that the expression levels of *IgA* and *IgG* had an increased tendency in acute and chronic cold stress. Su et al. [14,18] suggested that long-term and sustained cold stimulation increase the gene levels of immunoglobulin in ileum of broilers. Similar to previous studies, our results show a gradual increasing tendency in mRNA levels of *IgA* and *IgG* in the small intestine of broilers submitted to intermittent mild cold stimulation. *IgA* and *IgG* mRNA levels in C3 and C6 were higher than in CC, showing a significant increase in C3 as from day 22 of age. Collectively, our results indicate that exposure to a lower usual temperature might contribute to improving the intestinal immune function of broilers via stimulation of local immunoglobulin production. Our results were also consistent with previous reports by Hangalapura [28,30], in which has been demonstrated that broilers that had undergone a proper cold stimulation at an early age showed an enhanced immune function in late life.

Cytokines also play a key role in intestinal immunity under cold stimulation [10]. Th1 cells regulate and produce cytokines (most importantly *IL2* and *IFNγ*) to participate in cellular immunity, while T-helper 2 (Th2) cells mainly trigger *IL6* to mediate humoral immunity [35,36]. *IL8* activated neutrophils to clear pathogenic infections [37]. It has been reported that chronic cold stimulation upregulates mRNA expression levels of *IL2* and *IL6*, without significantly affecting *IFNγ* mRNA levels in chicken [38]. In human, cold stimulation was shown to increase serum *IL6* levels, especially at the initial phase of stimulation [39,40]. Hangalapura et al. [28] reported that cold stimulation can enhance Th1 related cellular immunity of chickens, consequently, increase the *IL2* and *IFNγ* mRNA levels. Additionally, the acute cold stimulation increased *IL8* levels in human lungs, and firstly increased then decreased the mRNA expression of proinflammatory cytokine *IL17* [41]. In this the present study, mRNA expression levels of intestinal *IL2* and *IFNγ* in C3 were higher than those in CC at the start of cold stimulation, which indicated that the broilers aimed to have a resistant capacity to the cold ambient by enhancing the Th1-mediated cellular immunity. At 43 days of age, mRNA expression of ileum *IL2* and *IFNγ* tended to stabilize. mRNA expression of *IL6* was slightly higher in duodenum C3 and C6 than in CC during cold stimulation treatment but did not show significant differences in levels detected in the jejunum and ileum after cold stimulation ceased. In addition, *IL6* mRNA levels in C3 group were significantly enriched compared to those in C6 and CC at 43 days of age, which might be an indication that broilers can suitably be managed at 3 °C below usual temperature.

TLRs are central components of the innate immune system and serve as the first line of host defense against invading microbial pathogens [42]. TLR signaling pathways are activated in response to temperature stress and lead to the release of endogenous molecules that serve as TLR ligands [43,44]. Heat stress can affect TLRs expression and modulate responsiveness of the innate and the adaptive immune system to fight pathogenic microorganisms [22]. *TLR2* is a low-affinity lipopolysaccharide (LPS) receptor, while *TLR4* transmits LPS signals [45]. Chronic cold stimulation triggers the overexpression of *TLR4* and indirectly stimulates hypothermia in humans [46]. *TLR5* can recognize bacterial flagellin [47]. Intracellular receptors *TLR7* and *TLR9* cooperate to distinguish host and exogenous nucleic acid-specific [48]. *TLR21* has a seemingly similar function in mammals [49]. When rats were infected necrotic enteritis and simultaneously exposed to acute cold stress at 4 °C for 10 min, mRNA expression levels of *TLR2* and *TLR4* in groups submitted to cold stress were significantly higher than those in control group, and the expressions in jejunum were lower than those in the ileum [50]. Basu et al. [51] found that with the reduction of cold stimulation temperature in 20, 15 and 10 °C, mRNA levels of *TLR2* were strongly increased then decreased equally to control group at 25 °C in the liver and kidney of catfish. mRNA level of *TLR4* increased with the decreased temperature of cold stimulating, and the mRNA level of *TLR5* was lower than that in control group [51]. Our results showed that *TLR2* mRNA levels in the duodenum and ileum of broilers in C3 were lower than in CC, while *TLR4* and *TLR2* mRNA levels were similar in the jejunum and ileum between C6 and CC. In contrast, no difference was observed in *TLR5* mRNA expression between CC group and cold stimulation group in small intestine. We speculated that the cold stimulus at 3 °C below the usual temperature for 3 and 6 h would slightly stimulate the body, which would in turn positively affect the immune defense function of the small intestine against the invasion of pathogenic microorganisms. There was no significant difference between control and cold stimulated groups regarding the mRNA expression of *TLR2*, *TLR4*, and *TLR5*. Interestingly, *TLR7* mRNA expression in C3 and C6 was significantly higher than in CC, while *TLR21* mRNA expression in C3 and C6 was significantly lower than in CC in the duodenum. Thus, long intermittent cold exposure ought to modulate the existing microbiome in the gut of broilers, for recognition of pathogen-associated molecular patterns to occur by *TLR7* and *TLR21* and enable the activation of immune cells in the small intestine.

## 5. Conclusions

The findings presented herein indicated that mild cold stimulation conditions at 3 °C below the usual temperature for 3 and 6 h every two days may have a positive impact on the intestinal immunity of broilers, as demonstrated by an upregulation of mRNA expression of intestinal immunoglobulins, cytokines, and TLRs.

## Figures and Tables

**Figure 1 animals-10-01492-f001:**
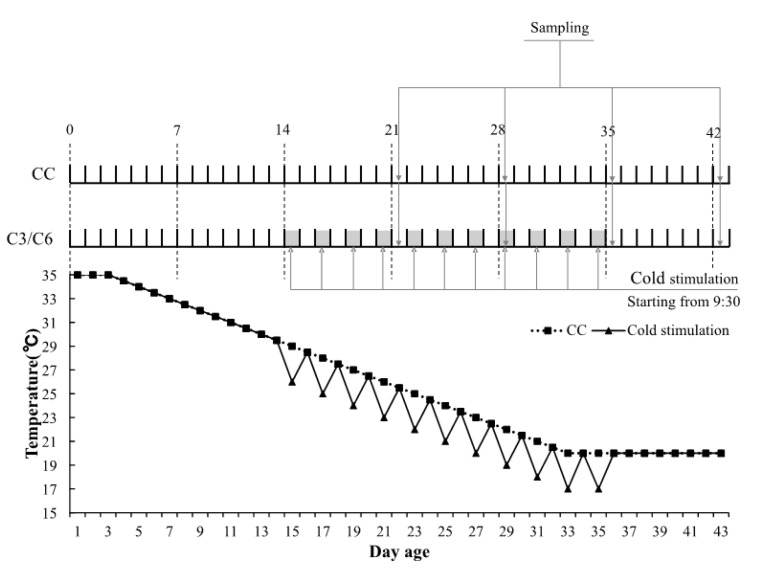
The temperature for experiment scheme.

**Figure 2 animals-10-01492-f002:**
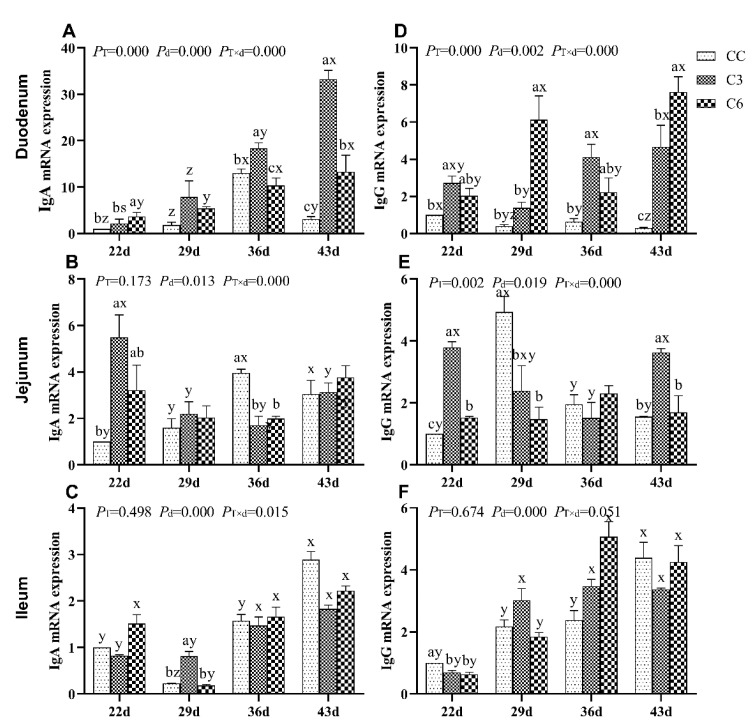
mRNA expression levels of immunoglobulins *IgA* and *IgG* in the duodenum (**A**,**D**), jejunum (**B**,**E**), and ileum (**C**,**F**) of broilers. Different letters indicate significant differences (*p <* 0.05) between treatment groups (a, b, c) and days of age (x, y, z, s).

**Figure 3 animals-10-01492-f003:**
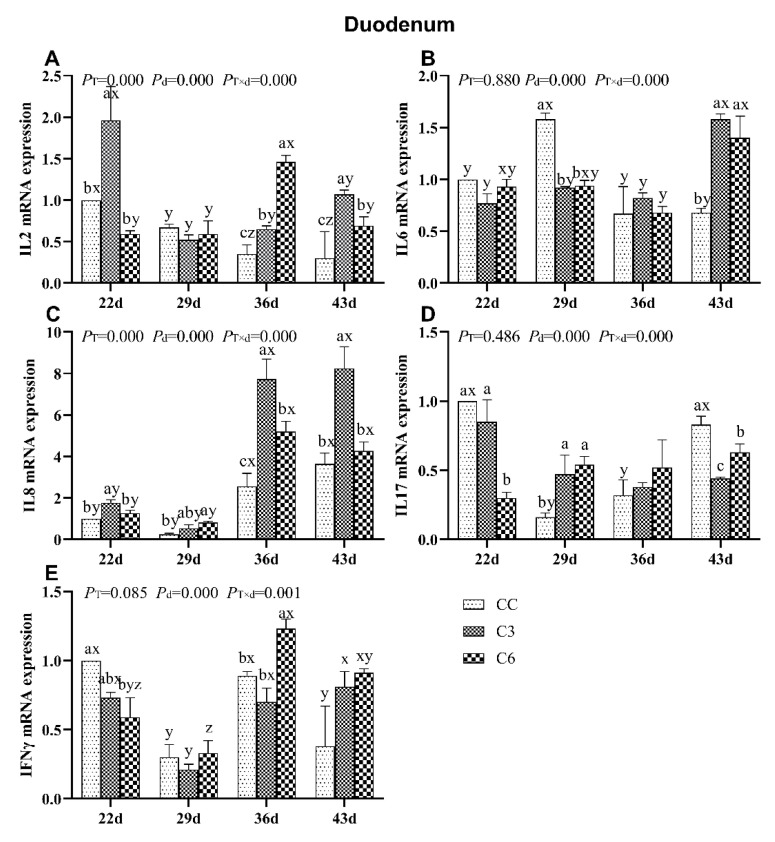
mRNA levels of cytokines *IL2* (**A**), *IL6* (**B**), *IL8* (**C**), *IL17* (**D**), and *IFNγ* (**E**) in the duodenum of broilers. Different letters indicate significant differences (*p <* 0.05) between treatment groups (a, b, c) and days of age (x, y, z).

**Figure 4 animals-10-01492-f004:**
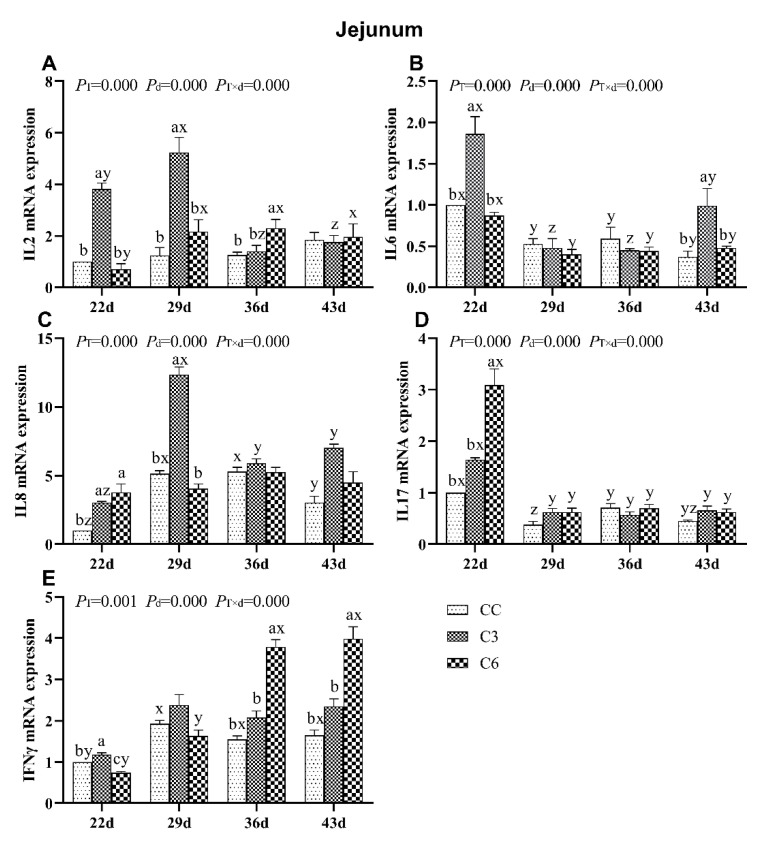
mRNA levels of cytokines *IL2* (**A**), *IL6* (**B**), *IL8* (**C**), *IL17* (**D**), and *IFNγ* (**E**) in the jejunum of broilers. Different letters indicate significant differences (*p <* 0.05) between treatment groups (a, b, c) and days of age (x, y, z).

**Figure 5 animals-10-01492-f005:**
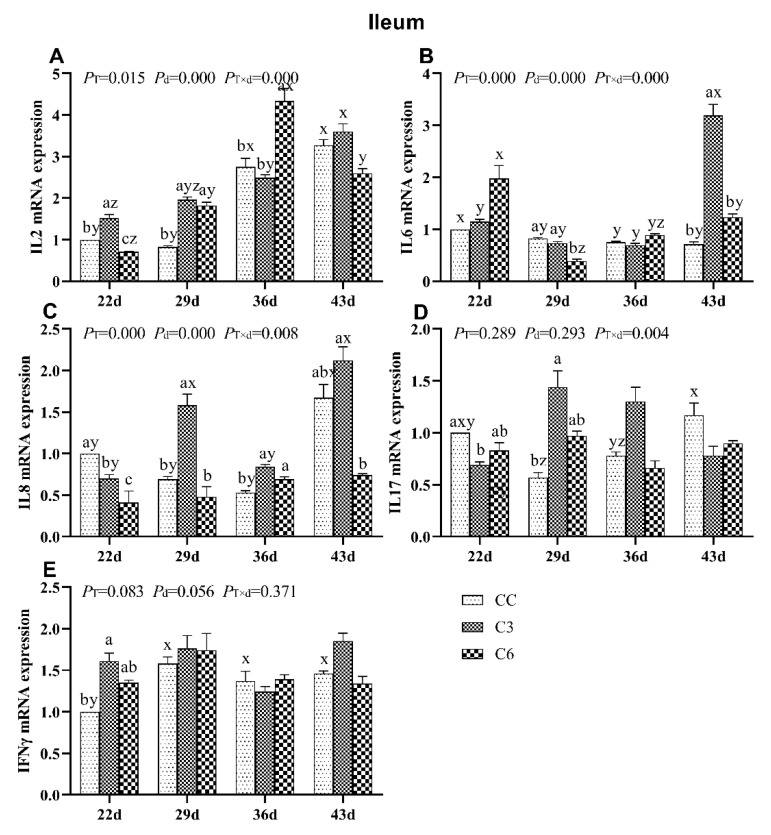
mRNA levels of cytokines *IL2* (**A**), *IL6* (**B**), *IL8* (**C**), *IL17* (**D**), and *IFNγ* (**E**) in the ileum of broilers. Different letters indicate significant differences (*p <* 0.05) between treatment groups (a, b, c, d) and days of age (x, y, z).

**Figure 6 animals-10-01492-f006:**
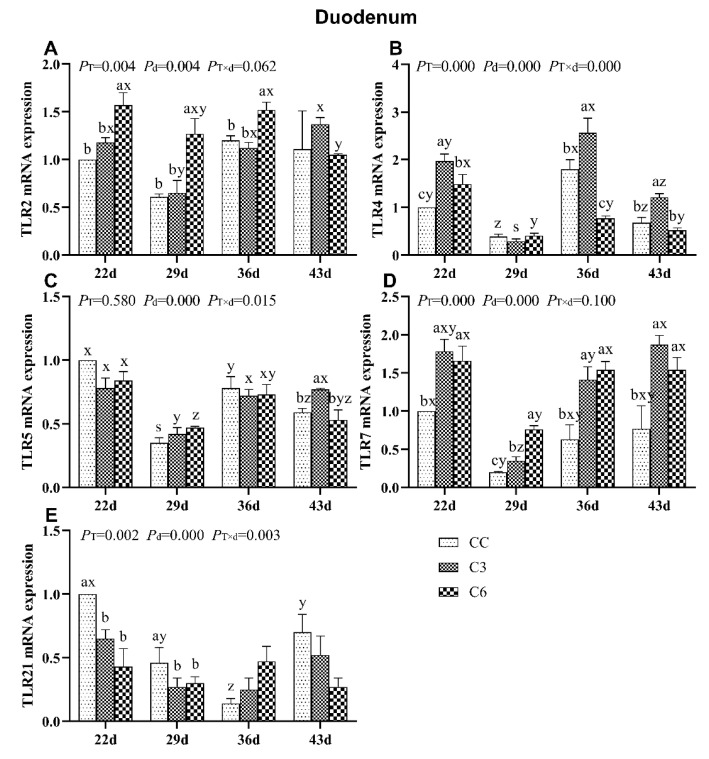
mRNA levels of Toll-like receptors *TLR2* (**A**), *TLR4* (**B**), *TLR5* (**C**), *TLR7* (**D**), and *TLR21* (**E**) in the duodenum of broilers. Different letters indicate significant differences (*p <* 0.05) between treatment groups (a, b, c, d) and days of age (x, y, z, s).

**Figure 7 animals-10-01492-f007:**
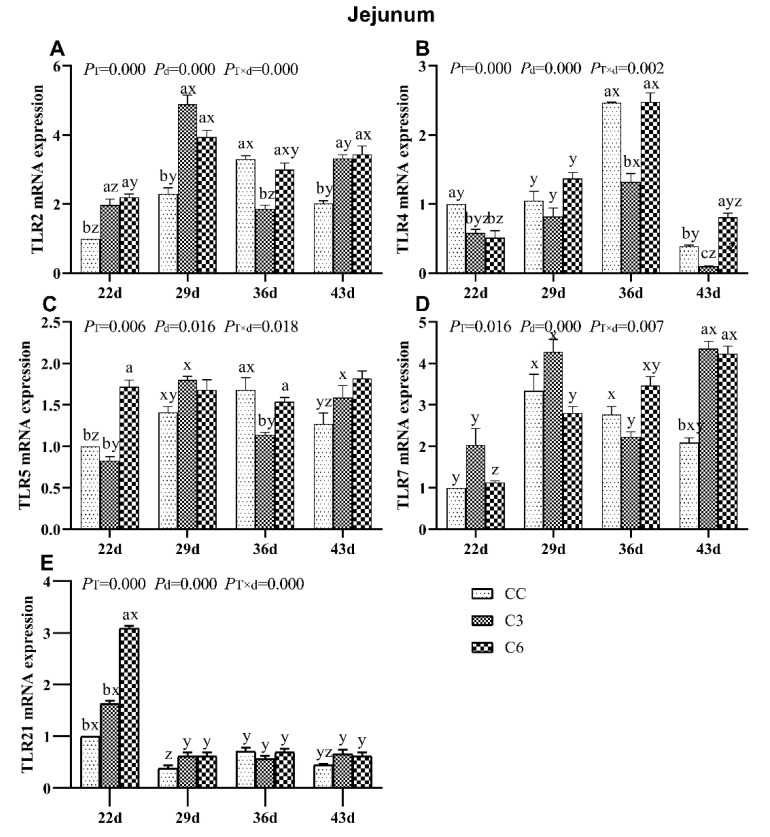
mRNA levels of Toll-like receptors *TLR2* (**A**), *TLR4* (**B**), *TLR5* (**C**), *TLR7* (**D**), and *TLR21* (**E**) in the jejunum of broilers. Different letters indicate significant differences (*p <* 0.05) between treatment groups (a, b, c) and days of age (x, y, z).

**Figure 8 animals-10-01492-f008:**
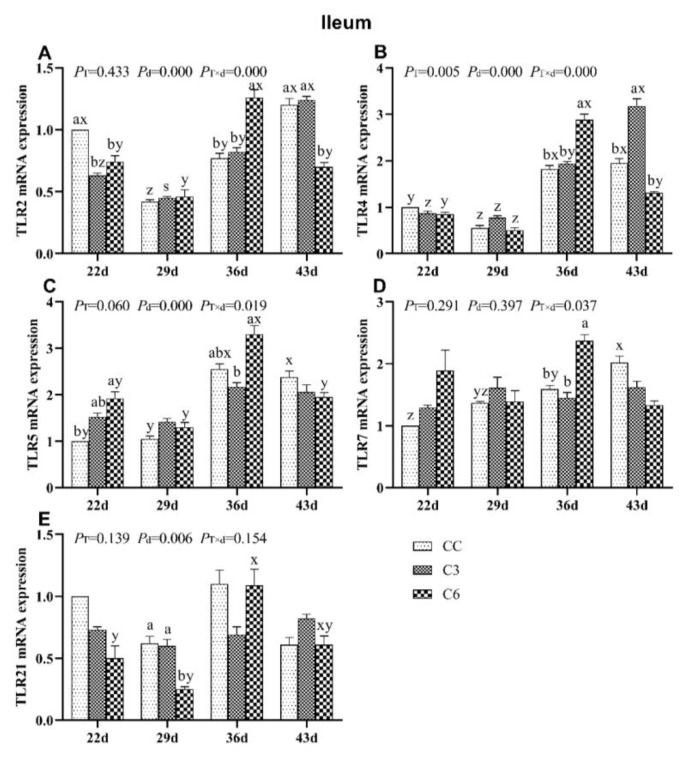
mRNA levels of Toll-like receptors *TLR2* (**A**), *TLR4* (**B**), *TLR5* (**C**), *TLR7* (**D**), and *TLR21* (**E**) in the ileum of broilers. Different letters indicate significant differences (*p <* 0.05) between treatment groups (a, b) and days of age (x, y, z, s).

**Table 1 animals-10-01492-t001:** Primer sequences used in the real-time quantitative reverse transcription PCR.

Gene	Reference Sequence	Primer Sequences (5′–3′)
*β-actin*	NM_205518.1	F: CACCACAGCCGAGAGAGAAAT
		R: TGACCATCAGGGAGTTCATAGC
*IgA*	NM_205287.1	F: TGCTAGTGGTTGTGGTGCTTGTG
		R: CGGAGGCGGAGGAGACGATG
*IgG*	XM_025146241.1	F: CGATTCCAGCCTCAGCGTCAC
		R: TAGGTGCCGTTGAAGTGTTCTTGG
*IFNγ*	NM_205149.1	F: GAACTGGACAGGGAGAAATGAGA
		R: ACGCCATCAGGAAGGTTGTT
*IL2*	NM_204153.1	F: CTGTATTTCGGTAGCAATG
		R: ACTCCTGGGTCTCAGTTG
*IL6*	NM_204628.1	F: AAATCCCTCCTCGCCAATCT
		R: CCCTCACGGTCTTCTCCATAAA
*IL8*	NM_205018.1	F: GGCTTGCTAGGGGAAATGA
		R: AGCTGACTCTGACTAGGAAACTGT
*IL17*	NM_204460.1	F: GCCATTCCAGGTGCGTGAACTC
		R: CGGCGGAGGACGAGGATCTC
*TLR2*	XM_001232192294a	F: GATTGTGGACAACATCATTGACTC
		R: AGAGCTGCTTTCAAGTTTTCCC
*TLR4*	NM_001030693.1190a	F: AGTCTGAAATTGCTGAGCTCAAAT
		R: GCGACGTTAAGCCATGGAAG
*TLR5*	NM_001024586124a	F: CCTTGTGCTTTGAGGAACGAGA
		R: CACCCATCTTTGAGAAACTGCC
*TLR7*	NM_001011688219b	F: TTCTGGCCACAGATGTGACC
		R: CCTTCAACTTGGCAGTGCAG
*TLR21*	NM_001030558112a	F: TGCCCCTCCCACTGCTGTCCACT
		R: AAAGGTGCCTTGACATCCT

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
