# Peer review of "Effects of Intermittent Mild Cold Stimulation on mRNA Expression of Immunoglobulins, Cytokines, and Toll-Like Receptors in the Small Intestine of Broilers"

_animals, 2020, doi:10.3390/ani10091492_

Round 1

Reviewer 1 Report

Figures 2 to 8 are still small to be clear enough, they contain too much information. Probably combining tables and figures, showing only comparisons with significant differences could be useful. 

Reviewer 2 Report

The authors have responded appropriately to resolve my concerns on the original manuscript. 

Reviewer 3 Report

In my opinion the manuscript can be published in Animals Journal with minor corrections.

I only have a few comments:

146-147 - Should be completed because according to Figure 2D a significant decrease in duodenal IgG mRNA expression was found also in C3 on day 36 compared to CC.

L: 186-187 - Should be completed because according to Figure 3D the IL17 mRNA level in C6 was also significantly higher on day 29 compared to CC group and in the case of C3 group the IL17 mRNA was significantly lower on day 43 and higher on 29 day compared to CC group.

L: 300-301 – Conversely, TLR2 mRNA levels in C6 showed significantly lower compared to CC on days 22 and 43, and higher on day 36. Moreover, TLR2 mRNA levels in C3 showed significantly lower compared to CC on days 22.

Author Response

This manuscript is a resubmission of an earlier submission. The following is a list of the peer review reports and author responses from that submission.

Round 1

Reviewer 1 Report

There are too many results (seven figures), then the Discussion section was a little bit short. It is suggested to divide results in two papers. 

Legends below figures do not need an ample explanation regarding materials and methods

There are some old references, they can be replaced by more recent ones

Reviewer 2 Report

The authors report a study on the effects of intermittant, short-term cold temperature exposure on expression of immune-related genes in broiler gastrointestinal tract tissues. 

The design and conduct of the study appear to be done appropriately. 

The discussion/conclusions seem to indicate that the authors think that higher expression of immune-related genes is always indicating better immunity, whereas it may be that some immune genes when expressed more highly can be detrimental. Because no measurement of inflammation or pathology was included in this study, it's not possible to say that higher levels of mRNA were always better for immune response.  Additionally, no measurement of the encoded proteins were made and the higher mRNA may not result in similarly high levels of proteins.

It would be more appropriate for the authors major message to be that they have strong evidence for modulation of immune response by cold temperature and that they suggest that these changes may result in positive impact on intestinal immunity.

Line 33. Meaning not clear -- please reword.

Lines 163-165.  This appears to be instructions -- delete. 

Figure 2. Explain the P value abbreviation.

Line 179. "remarkable fluctuation" is ambiguous -- please be more specific. 

Line 351. Should say "modulate" rather than "improve".

Line 395-396.  Meaning not clear -- please reword.

Line 413-416.  Meaning not clear -- does this mean that artificial microbiome should be established, or that the changes of cold temperature are expected to modulate the existing microbiome in the gut.

Refs #39, 40. Incorrect format. 

Reviewer 3 Report

The reviewed paper contains some original and interesting information about effects of intermittent mild cold stimulation on mRNA expression of immunoglobulins, cytokines, and Toll-like receptors in the small intestine of broilers. The title of the article is correct and is consistent with its content. The Lab methodology are correct. The article is interesting, but there are some points requirement specification in particular on the conclusions and applications of this work.

Generally, the study requires careful editing of the results obtained in accordance with what the authors themselves presented on the figures included in the work.

Detailed comments:

L: 141-143 -  Please reword this sentence. According to Figure 2A, a significant decrease in duodenal IgA mRNA expression was found in C6 on day 36 compared to CC.

L: 145-147 - Should be completed because according to Figure 2D a significant decrease in duodenal IgG mRNA expression was found also in C3 on day 36 compared to CC.

L: 155-156 – How they differed, more precisely - the level of IgG mRNA expression in C3 on day 43 was significantly higher compared to CC and C6.

L: 161-162 - This is not correct conclusion. Too generally.

Figure 2F - What about the data on the effect of cold stimulation on the level of IgG mRNA in C3 and C6 in ileum. According to Figure 2F a significant decrease in ileum IgG mRNA expression was found in C3 and C6 on day 22 compared to CC.

L: 186-187 - In my reading of Figure 3C, this is not correct conclusion, because according to Figure 3C IL8 mRNA level was higher in C6 on day 29 and 36, but not on day 43.

L: 190-192 – Too generally.

L: 207-209 - This is not correct conclusion. According to Figure 4A - as days progressed, IL2 mRNA levels in C3 decreased between days 29 to 36 (next it remains at the same level), and in C6 increased on day 29 and then it also remains at the same level on day 36 and 43. If the differences are not significant (in the case of  C6 on days 29, 36 and 43), then the expressions such as an increase or a decrease should be avoided.

L: 212 - In my reading of Figure 4C, this is not correct conclusion, because according to Figure 4C IL8 mRNA expression in C3 showed a significant increase compared to CC on days 22 and 29, but not 43.

L: 213 - This is not correct conclusion, because according to Figure 4C significant difference was found between CC and C6 on day 22.

L: 238 - This is not correct conclusion, because according to Figure 5B IL6 mRNA level decreased significantly in C6 on day 29 compared to CC.

L: 243-246 - This is not correct conclusion, because according to Figure 5E IFNγ mRNA expression was significantly enriched in C3 but only on day 22 (not on day 29) compared to CC.

L: 262-263 - In my reading of Figure 6A, this is not correct conclusion, because TLR2 mRNA expression in C3 decreased from days 22 to 29 and then increased on day 36, but not 43.

L: 264-266 - How they differed, more precisely.

L: 274-275 – however comparing to CC, TLR21 mRNA expression in C6 was significantly lower on days 22 and 29. The lack of significant differences between C3 and C6 does not mean that C6 is different from CC.

L: 297 - Should be completed because according to Figure 7C TLR5 mRNA level was significantly lower in C3 compared to C6 and CC on day 36.

L: 317 - Please provide more precisely what the differences.

L: 325 - This is not correct conclusion, because according to Figure 8C TLR5 mRNA level in C6 was higher than in C3 on day 36.
